# Post-consumer food waste generation while dining out: A close-up view

Myra Zeineddine[1], Samer Kharroubi[1], Ali Chalak[2], Hussein Hassan[3], Mohamad G. Abiad[1]*

1 Department of Nutrition and Food Sciences, Faculty of Agricultural and Food Sciences, American University of Beirut, Beirut, Lebanon, 2 Department of Agriculture, Faculty of Agricultural and Food Sciences, American University of Beirut, Beirut, Lebanon, 3 Nutrition Program, Department of Natural Sciences, School of Arts and Sciences, Lebanese American University, Chouran, Beirut, Lebanon

* ma192@aub.edu.lb

**Data Availability Statement:** Sharing our data on an open platform is restricted by University IRB guidelines and regulations. Data will be shared upon request due to University IRB requirements. Access to data requires the signature of an NDA.

## Abstract

Food loss and food waste occur along the food supply chain, negatively impacting the environment, global economy, and food security. There is a growing global interest in tackling this issue to mitigate or handle the waste generated and limit its repercussions, as one in eight people suffer from undernourishment worldwide. In the Arab world, where there is a high dependency on imports and limited potential of increasing local food production, addressing food loss and waste becomes substantial. Research has mainly been focused on household food waste generation, while data on post-consumer plate food waste in the foodservice sector remains scarce. In this study, managers from a representative sample of 222 restaurants located in Municipal Beirut, Lebanon, were surveyed about food waste generation. Plate food waste was measured to establish baseline information. Multiple Tobit regression analyses were performed to explore the determinants for plate food waste generation. Plate waste generation was also compared between Lebanese and non-Lebanese cuisine restaurants. Results revealed that 1,620 tons of plate food waste are generated per year in Beirut, equivalent to 0.15% of Lebanon's total organic waste. Furthermore, Lebanese cuisine restaurants serving Mediterranean Mezze were found to generate 34 kg of organic waste per day more than restaurants that serve international non-Lebanese cuisine. The type of cuisine, kind of service, and menu planning were significantly associated with post-consumer food waste generation. This study revealed an increasing concern towards the amount of plate waste generated in Beirut, and thereby further research is needed to create baseline information at the national level.

## Introduction

One-third of the overall food production is lost or wasted yearly worldwide, approximately equivalent to 1.3 billion tons of unconsumed foodstuffs along the whole food supply chain (FSC) [1–4]. According to the Food and Agriculture Organization of the United Nations [5], the aforementioned waste is worth around 1 trillion USD per year, raised to 2.6 trillion USD

Please contact irb@aub.edu.lb and the corresponding author at ma192@aub.edu.lb. The authors did not have special access privileges that others would not have when accessing the data except that the data will not have any identifiers (i.e., any link to the subjects). The data will be de-identified prior to sharing with IRB qualified interested researchers. Any researcher interested in the data can request access through the PI and approval will be sought accordingly. Conclusions can be reached by analyzing the de-identified data without the need for the raw data in specific.

**Funding:** The author(s) received no specific funding for this work.

**Competing interests:** The authors have declared that no competing interests exist.

when accounting for the environmental impacts' costs resulting from food waste. Additionally, one-eighth of the global population could be lifted from under-nourishment, given these amounts are not expended inattentively [6, 7]. It is indicated in an FAO study that 68% of the food is lost at the farmer level post-harvest, whereas 32% is wasted during consumption [2, 8].

The mitigation of the enormous amounts of food being lost or wasted is deemed necessary due to the subsequent negative impacts on the environment, economy, and food security [9–12]. Environmentally, food wastage results, among others, in an estimated annual carbon foot-print of 3.3 Gtons equivalents. Its disposal in landfills is a major emitter of methane gas ($CH_4$), 25 times more potent than carbon dioxide ($CO_2$) [13, 14] in terms of its contribution to climate change and global warming [15]. The water footprint–another major natural resource depletion resulting from food waste generation [2, 13], is estimated at 250 $km^3$ in water resources used for the agricultural production of wasted food [14], while the latter in turn occupies around 30% of the world's agricultural land area [16].

Economically, the negative impacts affect all the stakeholders along the FSC, whereby farmers losing the opportunity of higher income due to wasted investments and consumers face increased expenses [8, 17, 18], and often end up mitigating these effects by passing them on to consumers through higher prices [8, 19]. The yearly global economic loss resulting from food production never eaten is between 750 billion and 1 trillion USD [14], and 218 billion USD in the USA alone [20].

Food loss and waste (FLW) reduction and proper handling are crucial for food security attainment [8]. On account of the rising ethical predicaments, and while more than 850 million people remain undernourished worldwide, food waste needs to be addressed. It may contribute to the fight against hunger and malnourishment as reducing food waste can reduce retail prices, enhance affordability, and distribution, utilization, or further processing of the edible surplus [8, 21–23].

FLW studies in the literature mainly target developed countries. In contrast, studies in the Arab world (Middle East-North Africa and Near East-North Africa) remain scarce [22]. In the Arab region, it is estimated that, throughout the pre-consumption and consumption levels, an estimated 44% and 34% of the food is lost and wasted, respectively [24]. On a side note, with 40% of their agricultural commodities imported, these countries are highly dependent on food imports to meet their nutritional requirements [8, 22]. Yet food wastage in the region remains comparatively high; estimated at around 250 Kg per capita, it is higher than the global food waste average [8, 13, 15]. Reducing FLW generation is becoming more imperative to limit the challenges of safeguarding food security in the NENA region. The potential of increasing local food production remains restricted [8, 22, 25]. This being said, studies contributing to this effort in the NENA region shed light on the severity of household food waste in middle-income countries [26–30], while largely ignoring consumer food waste in the diners/hospitality industry.

As for Lebanon, published studies on food waste provide insights on household food wastage in terms of quantity; ethnographic effect among rural Lebanese communities; economic value–whereby approximately 5 to 10 USD are spent on food never eaten at home; association with nutrient loss–estimated at a caloric loss of 451 Kcal per day; purchasing behavior; beliefs, and attitudes driving food waste; in addition to consumer's perceived importance of waste and its mitigation [8, 10, 22, 28, 31, 32].

On the other hand, various local ongoing initiatives attempt to mitigate food waste. Such schemes include collecting unserved meals from food establishments and redistributing them to needy people or connecting food donors to those in need through food banks. Besides, other Lebanese initiatives raise awareness about the negative consequences of wasting food and testing waste management strategies to alleviate the magnitude of the environmental

burdens of landfilling through recycling, composting, among other solutions [13, 22, 32, 33]. Simultaneously, a lack of information in this area with a total absence of data focusing on post-consumer restaurant waste or plate food waste is still observed.

This study aims at quantifying the post-consumer food waste generated in a representative sample of restaurants in Beirut. It also aims to assess the determinants of this wastage and identify the factors related to it among the study sample while comparing Lebanese to non-Lebanese cuisine restaurants to investigate the differences in their post-consumer food waste patterns.

## Materials and methods

### Survey-based data collection

Referring to the Cadastral Districts of Municipal Beirut, administrative Beirut borders were drawn, and the areas within it were pinpointed. Based on the information available on Zomato Lebanon (www.zomato.com/beirut), a total of 1379 food establishments exist within the areas of interest, among which 514 are restaurants. Sample size calculations showed that a minimum of 221 restaurant managers based on a population size of 514 should be recruited to estimate a prevalence of 50% with a 95% confidence interval and a margin of error of 5%. Accounting for a refusal rate of 20%, 277 restaurants were approached. Only managers or head chefs have been considered eligible to take the survey since they are commonly in charge of the outlet's operations and better understand the outlets' purchasing and wasting activities. The study was reviewed and ethically approved by the Institutional Review Board (IRB) at the American University of Beirut (AUB) before conducting the study.

### Recruitment of participants

Using Zomato mobile application, outlets were located per area and street using the directions available for each restaurant's profile, which link the user to Google maps. Interested managers who met the eligibility criteria were consented. The data collection approach was conducted between January 2018 and May 2019 on the randomly selected restaurants in a private setting. Following written consent, eligible managers filled out a multicomponent questionnaire during a face-to-face interview session, which lasted between 20 to 30 minutes. Trained fieldworkers performed the interviews. The questionnaire was initially developed in English, subsequently translated to Arabic (since most of the managers talked Arabic), and then back-translated to English to verify the questionnaire's parallel-form reliability. Any disagreement that occurred in the back-translated version was resolved to provide a precise reading.

The questionnaire consisted of 42 questions. These relate to the restaurant characteristics, including the type of cuisine and kind of service offered, the organic and inorganic waste generated at the preparation and consumer levels, sorting and recycling, and managers' opinions on measures to mitigate and handle the food waste generated. The relevant questions within this project's scope related to the research inquiry on post-consumer food waste among Lebanese/Mediterranean and non-Lebanese restaurants were selected and analyzed. The remaining data will be potentially used for future work and projects of different aims and objectives.

### Statistical analysis

Data entry was completed on KOBO Toolbox using the survey format implemented on the application, accessed through a generated link. The results were exported to Microsoft Excel from the KOBO application. The data were then checked for completeness, and responses were coded and entered into the Statistical Package for the Social Sciences (SPSS) software

version 20 for Windows, which was later used for statistical analyses; IBM: Statistical Package for the Social Sciences (SPSS Statistics 2013). The data was further analyzed using Stata 16.1.

Descriptive statistics were presented to summarize the study variables of interest as counts and percentages for categorical variables and as means and standard deviations for the continuous ones. Censored Regression Analysis, i.e., Tobit Model, was applied to explore which factors are associated with the amount of food waste generated at the restaurants. The Tobit model is usually used to estimate linear relationships between variables when there is either left- or right-censoring in the dependent variable [34]. In our case, the dependent variable, food waste distribution, follows a mixed distribution where there is a probability mass at zero and a continuous distribution for values greater than zero. The nature of such dependent variable makes ordinary least squares potentially unsuitable, and the Tobit model offers a better estimation approach. Simple and multiple regression models were performed, using the food waste generated per day measured in Kg as the dependent variable whereas kind of service, managers' awareness about the negative consequences of food waste on the environment, economy, and society, importance of consumer behavior on amount of food waste generated, the effectiveness of menu planning on food waste reduction, the effectiveness of different portion size availability on food waste reduction, and challenges to reducing food waste, were all used as independent variables. All variables showing statistical significance in the simple regression model were included in the final multiple Tobit regression models as independent variables; this provides a way of adjusting or accounting for potentially confounding variables that have been included in the model. Results from regression analyses were expressed as a beta coefficient (β) with 95% confidence intervals (CI). For all analyses done, a p-value less than 0.05 was used to detect statistical significance.

## Results

Out of the 277 restaurants approached within administrative Beirut, 222 agreed to participate in this study, while the rest refrained from filling the survey due to lack of time or the manager's absence (80.14% response rate). The descriptive characteristics of the sample are summarized in Table 1.

### Post-consumer food waste quantification

Based on our findings, the average amount of organic waste generated and provided as a rough estimate by the managers was 20 kg per day, equivalent to 4.44 tons per day in Beirut or 1,620 tons per year. Findings also revealed that plate waste amounted to 54 kg/meal/restaurant/year or 12 tons/meal/year in Beirut, given the average number of meals served per day was estimated at 135 meals. At full restaurant occupancy, though, the average number of seats per outlet was estimated at 90 seats, resulting in 0.22 kg per capita per day or 81 kg of food wasted per person yearly.

### Post-consumer food waste determinants

**Characteristics of the study sample.** The study sample showed that 58 restaurants (26.1%) offered Lebanese/mezze-type cuisine, and 164 (73.9%) offered non-Lebanese cuisine. Restaurants serving non-mezze Mediterranean food, including American, Armenian, Asian, Chinese, French, German, Italian, Japanese, Mexican, South African, and Turkish cuisines, were all grouped under the variable "Non-Lebanese".

More than half of the managers approached reported working at restaurants that provide casual dining services (64.2%), while the remaining were either working at restaurants that provide fine dining services (18.8%), food on the go, or takeaway (11.9%), or self-service (5.1%).

**Table 1. Sample descriptive characteristics.**

| Characteristics (N = 222) | n (Valid %) |
|---|---|
| **Type of Cuisine** | |
| Lebanese | 58 (26.1) |
| Non-Lebanese | 164 (73.9) |
| **Kind of Service** | |
| Food on the go, takeaway | 21 (11.9) |
| Self-service | 9 (5.1) |
| Casual dining | 113 (64.2) |
| Fine dining (full service) | 33 (18.8) |
| **Awareness about the Negative Consequences of Food Waste on the Environment, Economy, and Society** | |
| Fully aware | 119 (55.9) |
| Somewhat aware | 57 (26.8) |
| Not very aware | 30 (14.1) |
| Totally unaware | 7 (3.3) |
| **Importance of Consumer Behavior on the Amount of Food Waste Generated** | |
| Most important | 132 (60.8) |
| Fairly important | 33 (15.2) |
| Important | 23 (10.6) |
| Slightly important | 10 (4.6) |
| Least important | 19 (8.8) |
| **Effectiveness of Menu Planning on Food Waste Reduction** | |
| Most effective | 100 (46.1) |
| Effective | 46 (21.2) |
| Neutral | 22 (10.1) |
| Somehow effective | 11 (5.1) |
| Not effective at all | 38 (17.5) |
| **Effectiveness of Different Portion Size Availability on Food Waste Reduction** | |
| Most effective | 68 (31.3) |
| Effective | 62 (28.6) |
| Neutral | 36 (16.6) |
| Somehow effective | 19 (8.8) |
| Not effective at all | 32 (14.7) |
| **Challenges to Reducing Food Waste** | |
| Lack of adequate food storage | 40 (19.1) |
| Food and Hygiene regulations | 55 (26.3) |
| Insufficient labor skills | 24 (11.5) |
| Lack of time | 34 (16.3) |
| Customer behavior | 56 (26.8) |

When asked about the level of awareness regarding the negative consequences of food waste on the environment, economy, and society, the majority of the surveyed managers (82.7%) indicated that they are aware of the problem, 55.9% of whom responded to be fully aware, and 26.8% were somewhat aware. On the other hand, only 14.1% of the respondents indicated that they are not very aware, whereas 3.3% were unaware of the negative consequences of food waste generation.

As for the question concerning the importance of the customer's behavior concerning the amount of food waste generated, 60.8% confirmed that it is the most important factor, whereas

the remaining believed it has a fairly important influence (15.2%), important (10.6%), least important (8.8%), and slightly important (4.6%).

With regards to the menu planning, 46.1% of the study participants believed that it is the most effective measure in post-consumer food waste reduction, while the remaining reported menu planning was effective (21.2%), not effective at all (17.5%), neutral opinion (10.1%), and only 5.1% reported menu planning was somehow effective at reducing the amount of post-consumer food waste generated at their outlets.

As for the effectiveness of the availability of different portion sizes on food waste reduction, 31.3% reported it was most effective, 28.6% responded as effective, 16.6% had a neutral stance, 14.7% thought it is not effective at all, and only 8.8% answered by somehow effective.

Among the five options given as challenges to reduce food waste, customer behavior was picked by 26.8% of the participants as the main challenge. To a lesser extent, food safety and hygiene regulations were the second main challenge, as indicated by 26.3% of the managers approached. On the other hand, 19.1% considered lack of adequate food storage as the main challenge at food waste reduction, and 16.3% considered lack of time primarily during rush hours as the main challenge, whereas only 11.5% attributed it to insufficient labor skills.

### Determinants of food waste generation

Table 2 displayed the results of the censored regression analysis. Simple and multiple regression models were performed to explore the effect of seven food waste determinants on the amount of post-consumer food waste generated per day in the selected restaurants within administrative Beirut. In the crude case, Lebanese cuisine ($\beta$ = 26.588, 95% CI 12.90 to 40.28), kind of service (fine dining: $\beta$ = 24.545, 95% CI 5.08 to 44.01), the effectiveness of menu planning (not effective at all: $\beta$ = 25.394, 95% CI 8.14 to 42.65) and effectiveness of different portion size (effective: $\beta$ = -17.404, 95% CI -33.68 to -1.13), were all significantly associated with food waste generation. After adjusting for covariates, Lebanese cuisine ($\beta$ = 34.068, 95% CI 18.70 to 49.43) and kind of service (fine dining: $\beta$ = 22.685, 95% CI 2.92 to 42.45) were also correlated with food waste generation, that is, restaurants that served Lebanese cuisine tend to have 34.1 kg of food waste, on average, generated per day more than restaurants that serve non-Lebanese cuisine while adjusting for kind of service. Furthermore, restaurants that provided fine dining food service tend to have 22.7 kg of food waste, on average, generated per day, more than restaurants that provide a casual dining service, while adjusting for Lebanese cuisine (Table 2).

The analysis also revealed that there is no statistically significant difference in the amount of post-consumer food waste generated, in both the crude and the multiple cases, among the following variables: awareness about the negative consequences of food waste on the environment, economy, and society, importance of consumer behavior on amount of food waste generated, and challenges to reducing food waste.

## Discussion

This study is the first national study to quantify the post-consumer food waste generation within the catering and hospitality industry, examine the determinants driving this wastage, and explore the difference in food waste quantities across different cuisines among a representative sample of foodservice establishments in administrative Beirut.

### Post-consumer food waste quantification

Quantifying food waste allows the creation of baseline information that, in turn, enables detecting, measuring, and assessing change over time. Given the absence of data related to

**Table 2. Determinants of food waste generation–censored regression analysis.**

| | | Crude Case (Single Regression Analysis) | Multiple Case (Multiple Regression Analysis) |
|---|---|---|---|
| **Type of Cuisine** | Non-Lebanese | Reference | Reference |
| | Lebanese | **26.588 (12.90, 40.28), p<0.0001** | **34.068 (18.70, 49.43), p<0.0001** |
| **Kind of Service** | Casual dining | Reference | Reference |
| | Takeaway | -0.085 (-22.04, 21.87), p = 0.994 | 0.737 (-21.82, 23.30), p = 0.949 |
| | Self-service | -11.307 (-42.41, 19.80), p = 0.474 | -6.877 (-37.04, 23.28), p = 0.653 |
| | Fine dining | **24.545 (5.08, 44.01), p = 0.014** | **22.685 (2.92, 42.45), p = 0.025** |
| **Awareness about the Negative Consequences of Food Waste on the Environment, Economy, and Society** | Fully aware | Reference | Reference |
| | Somewhat aware | 4.595 (-10.51, 19.70), p = 0.549 | |
| | Not very aware | -5.312 (-23.60, 12.97), p = 0.567 | |
| | Totally unaware | -12.445 (-46.40, 21.51), p = 0.47 | |
| **Importance of Consumer Behavior on the Amount of Food Waste Generated** | Most important | Reference | Reference |
| | Fairly important | -1.735 (-20.12, 16.65), p = 0.852 | |
| | Important | -8.011 (-28.31, 12.29), p = 0.437 | |
| | Slightly important | -7.457 (-36.21, 21.30), p = 0.61 | |
| | Least important | -6.342 (-31.03, 18.35), p = 0.613 | |
| **Effectiveness of Menu Planning on Food Waste Reduction** | Most effective | Reference | Reference |
| | Effective | 0.226 (-15.81, 16.26), p = 0.978 | 5.944 (-12.72, 24.61), p = 0.53 |
| | Neutral | -4.015 (-25.60, 17.57), p = 0.714 | -0.708 (-25.60, 24.18), p = 0.955 |
| | Somehow effective | 2.623 (-27.21, 32.46), p = 0.863 | 5.923 (-26.67, 38.51), p = 0.72 |
| | Not effective at all | **25.394 (8.14, 42.65), p = 0.004** | 18.286 (-2.17, 38.74), p = 0.079 |
| **Effectiveness of Different Portion Size Availability on Food Waste Reduction** | Most effective | Reference | Reference |
| | Effective | **-17.404 (-33.68, -1.13), p = 0.036** | -14.404 (-32.63, 3.83), p = 0.12 |
| | Neutral | -17.068 (-35.85, 1.72), p = 0.075 | -13.838 (-36.25, 8.58), p = 0.224 |
| | Somehow effective | -15.842 (-39.22, 7.54), p = 0.183 | -16.436 (-40.91, 8.04), p = 0.186 |
| | Not effective at all | -14.086 (-34.58, 6.41), p = 0.177 | -15.888 (-39.32, 7.54), p = 0.182 |
| **Challenges to Reduce Food Waste** | Customer behavior | Reference | Reference |
| | Lack of adequate food storage | -2.360 (-22.42, 17.70), p = 0.817 | |
| | Food Safety and Hygiene regulations | 11.371 (-5.92, 28.66), p = 0.196 | |

*(Continued)*

**Table 2.** (Continued)

|  |  | Crude Case (Single Regression Analysis) | Multiple Case (Multiple Regression Analysis) |
|---|---|---|---|
|  | Lack of time | 12.650 (-7.21, 32.51), p = 0.21 |  |
|  | Insufficient labor skills | 11.849 (-10.38, 34.08), p = 0.294 |  |

food waste generated by food establishments in the NENA region in general and specifically in an urban setting such as Beirut, this project is a great addition to the literature.

This study revealed that the post-consumer food waste generated in restaurants in Beirut is equal to 1,620 tons per year, equivalent to 0.15% of Lebanon's total organic wastage generated yearly. This amount is smaller than the plate waste quantities observed in the US, where 16 million tons of waste from the foodservice sector are generated per year, in Europe where 12.5 million tons are generated per year, and in European countries, namely Sweden (99 kilotons per year), Germany (1.9 million tons per year), and France (2 million tons per year) [20, 35, 36]. This can be attributed to differences in country size, population size, restaurant capacity, income, and culture, noting that our results are not representative on the national level. Additionally, our results revealed an 81 kg/cap/year of food waste generation as compared to 210 kg/cap/year in the Arab world, 95–115 kg/cap/year in Europe and North America, 139 kg/cap/year in Singapore, and 170 kg/cap/year in South Africa [22].

However, for a small country like Lebanon, given the examined sample represents the capital only, such high levels are alarming. They are also more worrisome as Lebanon highly depends on food imports, similar to all other countries in the Arab region, with limited resources for increased food production to sustain food and nutrition security [22, 37].

### Post-consumer food waste determinants

Concerning post-consumer food waste drivers, the results of this study showed that the type of cuisine served at the food establishment, the kind of service provided, and menu planning affect the quantity of plate waste generated. One of our unique findings is that consumers tend to waste more when dining out at Lebanese cuisine restaurants versus other types of cuisines, with an average of 34 kg of post-consumer food waste per day more than restaurants that serve non-Lebanese cuisine. The latter can be explained by the wide variety of dishes available under the Lebanese cuisine, served in average quantities and meant to be shared, apart from the main course options. These starter dishes are known as mezze dishes. Consumers dining out at Lebanese cuisine restaurants end up over-ordering mezze dishes to share, which is in turn part of the culture, along with one or many main course meals. This leads to a higher chance of plate waste generation than other cuisines that serve one meal portion per person, where the chance of wasted left-overs from over-ordering and sharing is much less. Additionally, over-ordering is also impacted by the cultural aspect and the Lebanese people's generosity of honoring the guest, linked to having generous amounts of food with a wider variety of dishes and offering a larger quantity than needed, matching the number of individuals at the table. There are no comparative studies between post-consumer food waste generated between Lebanese and non-Lebanese cuisine restaurants to the best of our knowledge.

Moreover, restaurants that provide fine dining services tend to generate, on average, around 23 kg of post-consumer food waste per day more than the ones that offer casual dining services based on our results. This finding is consistent with the previous research, which

showed that fine diners in the US generate a difference of 3 million tons of plate waste per year more than casual diners [20]. A study from Shanghai, China, interviewed managers and chefs from the fine dining foodservice sector who reported that this kind of service causes not only excessive waste generation at the cooking stage for aesthetic purposes but also at the consumer level [38]. Besides, plate food waste at fine diners is potentially caused by the greediness of fine dining restaurant guests who commonly worry less about their expenses and end up wasting more due to over-ordering; which in turn is encouraged because of the extensive menus developed to enhance customer satisfaction and to stand out from competitors [35, 39, 40]. This can be further explained given the traditions of our culture, where Lebanese people over-order out of generosity as a welcoming gesture.

Menu planning has been proven effective at reducing the plate waste generated based on our results. Restaurants, where managers reported that menu planning is not effective at all at post-consumer food waste reduction, happened to have on average 18 Kg of food waste generated per day more than the ones where managers reported this measure to be the most effective for food waste reduction. This is in line with the literature and can be explained as long menus with a broad range of choices push customers to order more even when they know they won't be able to finish their meals, leading to more food waste [1, 35, 41, 42]. The effect of menu size is further amplified in restaurants offering a Lebanese type of cuisine, where menus are diverse with a wide variety of food options. In such cuisines, customers request an excessive number of dishes for wider meal combinations, especially that mezze dishes are served in sizeable portions and a wider variety. Add to that the cultural effect and Arab generosity, which tops it all up.

## Conclusion and recommendations

The post-consumer food waste reduction in Beirut, given the alarming amount of 1,620 tons generated per year, is of major relevance. The diversity and broad range of options available in Lebanese cuisine and the people's cultural generosity lead to higher plate waste generation caused primarily by over-ordering mezze dishes to share. Fine diners happened to collect larger quantities of plate waste post-consumer than casual diners, which is linked to the generosity and higher income of fine dining food service guests who tend to worry less about their expenses. Last but not least, menu planning was found to be effective at reducing food waste since long menus confuse customers and most likely push them to over-order for the sake of trying more items, which in turn contributes to waste generation further.

Ongoing initiatives in Lebanon continue to help handle the food waste generated by collecting and processing or redistributing it to people in need. Local awareness campaigns also happen actively to highlight the negative consequences of wasting on food security, the environment, and the economy. Additionally, local testing and improvement of waste management strategies such as recycling, composting, and waste conversion to animal feed alleviate the burden of landfilling on the environment, especially since Lebanon is a small country with limited landfilling spaces.

Further research is required to collect nation-wide representative data and create baseline information about the quantities of food waste generated to enable detecting and measuring change over time. It is also important to assess food waste determinants across different country regions where people come from different backgrounds, religions, education levels, socioeconomic class, and cultures. Local authorities' support is needed to provide financial resources to conduct the research, personnel to collect the data, and qualified individuals to train restaurant managers and launch awareness campaigns to limit the quantities of food waste generated at the food establishment level.

### Proposed policy against plate food waste

**Syndicate of restaurants, cafes, night clubs, and pastries in Lebanon.**

- Develop, finalize, and publish the draft of the law on food waste proposed in 2019 with GWR Consulting [43].

- Encourage restaurant owners to become members of the syndicate to have a bigger impact, as it advocates on behalf of the Food and Beverage industry and acts as a lobbying body on the governmental level.

- Push for governmental financial support or lobby for funds and grants to conduct the necessary research on the national level.

- Measure and detect a change in the quantities of post-consumer food waste generation over time after creating baseline information.

- Collaborate with food safety private sector or local authorities–such as the Ministry of Public Health, to create a team in charge of providing training for left-overs' food safety assessment for donations.

- Provide training for restaurant owners and managers to limit preparation and plate food waste generation through crew training.

- Place fridges and stations, promote food excess or left-over deposit for public donations and collaborate with municipalities to clean and maintain [44].

- Raise awareness of the negative impacts of food waste using ads on TV, radio, and social media platforms.

- Collaborate with online local food ordering applications to incorporate the number of individuals as a factor in the amount of food selected to limit over-ordering.

- Audit restaurant managers and service area crew through mystery shoppers to assess their behavior towards over-ordering encounters and left-overs takeaway encouragement.

**Restaurant owners and managers.**

- Provide training and continuous reinforcement for restaurant service area crew to notify customers in the case of over-ordering.

- Plan focused menus to limit over-ordering behaviors encouraged by the availability of a broad range of choices.

- Promote and encourage left-overs takeaway.

- Offer different portion sizes and offer smaller plates to limit food waste generation [45].

- Identify which meals generate the most left-overs and eventually decrease their quantity.

- Suggest offering half of the plate portion for dine-in and immediate to-go packaging of the remaining half portion.

- Offer mezze dishes tray in Lebanese cuisine restaurants with smaller than average portion size meals for sharing and limited waste generation.

- Set a punitive law that charges both managers and customers with fines and penalties for unfinished plates at restaurants due to over-ordering in case of discarding left-overs instead of packing them for carry-out [45, 46].

- Train kitchen-dedicated personnel for safe sorting and excess food separation for donations.

- Collaborate with local initiatives and organizations that collect food waste and make it accessible to food insecure people and refugees.

### Local municipalities.

- Promote, reinforce, and incentivize sorting.

- Collaborate with local authorities to recycle.

- Organize campaigns to raise awareness of the negative consequences of food waste through informative messages and reminders.

- Oblige restaurants to contract with local charity organizations to redistribute safe left-over food items to the people in need [44].

## Author Contributions

**Conceptualization:** Mohamad G. Abiad.

**Formal analysis:** Myra Zeineddine, Samer Kharroubi, Ali Chalak, Mohamad G. Abiad.

**Investigation:** Myra Zeineddine.

**Methodology:** Ali Chalak, Mohamad G. Abiad.

**Project administration:** Mohamad G. Abiad.

**Supervision:** Samer Kharroubi, Mohamad G. Abiad.

**Writing – original draft:** Myra Zeineddine.

**Writing – review & editing:** Samer Kharroubi, Ali Chalak, Hussein Hassan, Mohamad G. Abiad.

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
