## [Decision Letter · Decision Letter 0]

7 Apr 2021

PONE-D-21-07906

Post-Consumer Food Waste Generation while Dining Out: A Close-up View

PLOS ONE

Dear Dr. MOHAMAD G. ABIAD,

Thank you for submitting your manuscript to PLOS ONE. After careful consideration, we feel that it has merit but does not fully meet PLOS ONE’s publication criteria as it currently stands. Therefore, we invite you to submit a revised version of the manuscript that addresses the points raised during the review process.

We look forward to receiving your revised manuscript.

Kind regards,

László VASA, PhD

Academic Editor

PLOS ONE

Journal Requirements:

Once you have amended this statement in the Methods section of the manuscript, please add the same text to the “Ethics Statement” field of the submission form (via “Edit Submission”).

Reviewers' comments:

Reviewer's Responses to Questions

**Comments to the Author**

1. Is the manuscript technically sound, and do the data support the conclusions?

Reviewer #1: Partly

Reviewer #2: Yes

2. Has the statistical analysis been performed appropriately and rigorously? 

Reviewer #1: Yes

Reviewer #2: I Don't Know

3. Have the authors made all data underlying the findings in their manuscript fully available?

Reviewer #1: No

Reviewer #2: No

4. Is the manuscript presented in an intelligible fashion and written in standard English?

Reviewer #1: Yes

Reviewer #2: Yes

5. Review Comments to the Author

Reviewer #1: The paper investigates the very actual problem of food waste generation.

Tha authors used the appropriate methodological toolset, performed the research well and the conclusions are right.

However, the literature review is missing from the articel, a separate chapter should be created for it and also more souces should be reviewed.

Reviewer #2: The study presented in this article is of outstanding scientific and practical importance.

The authors did significant background work. Data collection and processing required tremendous work.

I see a shortcoming in the description of the chosen statistical method. Why was the Tobit model chosen? What are the advantages and disadvantages compared to other methods? Eg.: https://onlinelibrary.wiley.com/doi/full/10.1002/gsj.1363

I think it’s conceivable that this is the best choice, but you should justify why.

Without knowing the basic data, I cannot judge the quality of statistical analysis.

It is a less important issue, but the figures should be re-edited. It would be worthwhile to think about what should be shown in a figure and what can be clearly seen in a table.

6. PLOS authors have the option to publish the peer review history of their article (what does this mean?). If published, this will include your full peer review and any attached files.

Reviewer #1: No

Reviewer #2: No

---

## [Author Response · Author response to Decision Letter 0]

3 May 2021

PONE-D-21-07906

Post-Consumer Food Waste Generation while Dining Out: A Close-up View

PLOS ONE

Dear Dr. László VASA,

Thank you very much for sending us the reviews of our manuscript. We mostly agree with the reviewers' observations and tried to address the comments and concerns detailed below. However, sharing our data on an open platform is restricted by University IRB guidelines and regulations.

Data will be shared upon request due to University IRB requirements. Access to data requires the signature of an NDA. Please contact irb@aub.edu.lb and the corresponding author at ma192@aub.edu.lb. The authors did not have special access privileges that others would not have when accessing the data except that the data will not have any identifiers (i.e., any link to the subjects).

On another note, on behalf of the authors, I declare that no competing interests exist.

We want to thank the reviewers for raising questions whose answers improved the quality of our manuscript. 

Please find here below our responses to the reviewers' comments/suggestions.

Best regards,

Mohamad Abiad

Journal Requirements:

When submitting your revision, please address these additional requirements.

2. Please provide additional details regarding participant consent. In the ethics statement in the Methods and online submission information, please ensure that you have specified what type you obtained (for instance, written or verbal, and if verbal, how it was documented and witnessed). 

The consent was written. We have edited the text in the revised manuscript. 

Once you have amended this statement in the Methods section of the manuscript, please add the same text to the "Ethics Statement" field of the submission form (via "Edit Submission").

This has been addressed in the cover letter.

Data will be shared upon request due to University IRB requirements. Access to data requires the signature of an NDA. Please contact irb@aub.edu.lb and the corresponding Author at ma192@aub.edu.lb. The authors did not have special access privileges that others would not have when accessing the data except that the data will not have any identifiers (i.e., any link to the subjects).

Reviewers' comments:

Reviewer's Responses to Questions

Comments to the Author

1. Is the manuscript technically sound, and do the data support the conclusions?

Reviewer #1: Partly

Reviewer #2: Yes

We thank the reviewers for their feedback.

2. Has the statistical analysis been performed appropriately and rigorously?

Reviewer #1: Yes

Reviewer #2: I Don't Know

We believe that we have run rigorous statistical analysis on the collected data. 

3. Have the authors made all data underlying the findings in their manuscript fully available?

Reviewer #1: No

Reviewer #2: No

we have highlighted the reason why the data has not been made available.

Data will be shared upon request due to University IRB requirements. Access to data requires the signature of an NDA. Please contact irb@aub.edu.lb and the corresponding Author at ma192@aub.edu.lb. The authors did not have special access privileges that others would not have when accessing the data except that the data will not have any identifiers (i.e., any link to the subjects).

4. Is the manuscript presented in an intelligible fashion and written in standard English?

Reviewer #1: Yes

Reviewer #2: Yes

We thank the authors for their positive feedback. 

Reviewer #1: 

The paper investigates the very actual problem of food waste generation.

The authors used the appropriate methodological toolset, performed the research well, and the conclusions are right.

However, the literature review is missing from the article, a separate chapter should be created for it, and more sources should be reviewed.

We thank the reviewer for his suggestion, but we disagree that more sources need to be reviewed. We believe we have done a rigorous search and included all relevant literature on the subject. The second reviewer acknowledges this "the authors did significant background work." 

Reviewer #2: 

The study presented in this article is of outstanding scientific and practical importance.

The authors did significant background work. Data collection and processing required tremendous work.

I see a shortcoming in the description of the chosen statistical method. Why was the Tobit model chosen? What are the advantages and disadvantages compared to other methods? Eg.: https://onlinelibrary.wiley.com/doi/full/10.1002/gsj.1363

I think it's conceivable that this is the best choice, but you should justify why.

The Tobit model is usually used to estimate linear relationships between variables when there is either left- or right-censoring in the dependent variable. In our case, the dependent variable is assumed to follow a mixed distribution where there is a probability mass at zero and a continuous distribution for values greater than zero. The zeros and the non‐zero values thus come from the same data generating process (a data set typically called corner solution). The nature of such dependent variable makes ordinary least squares (OLS, henceforth) potentially unsuitable, and Tobit models may constitute a valid estimation approach. 

Without knowing the basic data, I cannot judge the quality of statistical analysis.

It is a less important issue, but the figures should be re-edited. It would be worthwhile to think about what should be shown in a figure and what can be clearly seen in a table.

We thank the reviewer. We have included the data explanation in text.

---

## [Editor Report · Decision Letter 1]

6 May 2021

Post-Consumer Food Waste Generation while Dining Out: A Close-up View

PONE-D-21-07906R1

Dear Dr. MOHAMAD G. ABIAD,

We’re pleased to inform you that your manuscript has been judged scientifically suitable for publication and will be formally accepted for publication once it meets all outstanding technical requirements.

Kind regards,

László VASA, PhD

Academic Editor

PLOS ONE
---

## [Editor Report · Acceptance letter]

12 May 2021

PONE-D-21-07906R1 

Post-Consumer Food Waste Generation while Dining Out: A Close-up View 

Dear Dr. Abiad:

I'm pleased to inform you that your manuscript has been deemed suitable for publication in PLOS ONE. Congratulations! Your manuscript is now with our production department. 

Kind regards, 

on behalf of

Prof. Dr. László VASA 

Academic Editor

PLOS ONE